# The Clever Hans Mirage: A Comprehensive Survey on Spurious Correlations in Machine Learning

**Wenqian Ye**[1]*   **Luyang Jiang**[2]*   **Eric Xie**[1]*   **Guangtao Zheng**[1]
**Yunsheng Ma**[2]   **Xu Cao**[3]   **Dongliang Guo**[1]   **Daiqing Qi**[1]   **Zeyu He**[4]   **Yijun Tian**[5]
**Megan Coffee**[6]   **Zhe Zeng**[1]   **Sheng Li**[1]   **Ting-Hao 'Kenneth' Huang**[4]   **Ziran Wang**[2]
**James M. Rehg**[3]   **Henry Kautz**[1]   **Aidong Zhang**[1]

[1]*University of Virginia*   [2]*Purdue University*   [3]*University of Illinois Urbana-Champaign*
[4]*Pennsylvania State University*   [5]*Amazon*   [6]*New York University*

**Reviewed on OpenReview:** *https://openreview.net/forum?id=kIuqPmS1b1*

## Abstract

Back in the early 20th century, a famous horse named Hans appeared to perform arithmetic and other intellectual tasks during exhibitions in Germany, which has drew widespread attention. However, later studies showed that Hans relied solely on subtle, involuntary cues in the trainer's body language. Modern machine learning models are no different. These models are known to be sensitive to *spurious correlations* between non-essential features of the inputs (e.g., background, texture, and secondary objects) and the corresponding prediction labels. Such features and their correlations with the labels are known as "spurious" because they tend to change with shifts in real-world data distributions, which can negatively impact the model's generalization and robustness. In this survey, we provide a comprehensive survey of this emerging issue, along with a fine-grained taxonomy of existing state-of-the-art methods for addressing spurious correlations in machine learning models. Additionally, we summarize existing datasets, benchmarks, and metrics to facilitate future research. The paper concludes with a discussion of the broader impacts, the recent advancements, and future challenges in the era of Generative Artificial Intelligence (GenAI), aiming to provide valuable insights for researchers in the related domains of the machine learning community.

## 1 Introduction

While machine learning (ML) systems have made remarkable strides, distribution shifts and adversarial robustness have become critical issues in the past decade (Taori et al., 2020). It is increasingly evident that one of the major causes behind these issues is the strong reliance on spurious correlations between superficial features of the inputs (e.g., background, texture, and secondary objects) and the corresponding labels. When these correlations captured during training no longer hold in the test data, the performance of ML models deteriorates, leading to robustness issues and negative social impact in critical domains such as healthcare. For instance, studies in pneumonia detection with Convolutional Neural Networks (CNNs) have shown that models often depend on extraneous cues, like metal tokens in chest X-ray radiographs from different hospitals, instead of learning the actual pathological features of the disease (Kirichenko et al., 2023; Zech et al., 2018). Therefore, addressing spurious correlations is of vital importance.

In recent years, spurious correlations have been referred to in various terms, such as shortcut learning, group robustness, and simplicity bias. Significant progress has been made in both detection and mitigation of these phenomena in fields such as computer vision (Wang et al., 2021; Yang et al., 2022b), natural language processing (Du et al., 2022b), and healthcare (Huang et al., 2022). Despite these advances, a comprehensive survey that formally defines spurious correlations and systematically reviews associated learning algorithms and challenges has yet to be published.

---

*Core contributors.

In this paper, we present the first comprehensive survey on spurious correlations, providing the formal definitions, a taxonomy of current state-of-the-art methods for addressing spurious correlations in machine learning models, and an overview of relevant datasets, benchmarks, and evaluation metrics. We further discuss future research challenges and explore the potential role of foundation models in mitigating the adverse effects of spurious correlations. We hope this survey serves as a comprehensive reference and inspires further research in robust machine learning.

## 2   Spurious Correlations

In statistics, spurious correlation, namely "*correlations that do not imply causation*," refers to two random variables that appear related to each other, but their true relationship is either coincidental or confounded by an external variable. This phenomenon leads to misleading or incorrect interpretations of data and models.

A classic historical analogy is *Clever Hans* (Figure 1), a horse in the early 20th-century Germany that drew worldwide attention for performing arithmetic and other intellectual tasks (Lapuschkin et al., 2019). However, Hans did not actually understand these concepts. Instead, it responded based on subtle, unintentional cues in his trainer's posture. Today, machine learning models/agents tend to exhibit similar behavior, relying on non-semantic, spurious features that correlate with the correct answer during training, but fail to generalize to broader or shifted distributions. We first give a formal definition based on previous literature.

**Definition 2.1** (Spurious Correlation)**.** Let $\mathcal{D}_{tr} = \{(x_i, y_i)\}_{i=1}^n$ be the training set with $x_i \in \mathcal{X}$ and $y_i \in \mathcal{Y}$, where $\mathcal{X}$ denotes the set of all possible inputs and $\mathcal{Y}$ denotes the set of $K$ classes. For each data point $x_i$ with the class label $y_i$, there exists a spurious attribute $a_i \in \mathcal{A}$ that is non-predictive of $y_i$, where $\mathcal{A}$ denotes the set of all possible spurious attributes. A spurious correlation, denoted $\langle y, a \rangle$, is the association between $y \in \mathcal{Y}$ and $a \in \mathcal{A}$, where $y$ and $a$ exist in a one-to-many mapping $\phi : \mathcal{A} \mapsto \mathcal{Y}^{K'}$ conditioned on $\mathcal{D}_{tr}$ (with $1 < K' \leq K$), where $\mathcal{Y}^{K'}$ denotes the set of $K'$ labels. A set of data exhibiting the spurious correlation $\langle y, a \rangle$ is annotated with the group label $g = (y, a)$, where $\mathcal{G} \coloneqq \mathcal{Y} \times \mathcal{A}$ is the set of combinations of class labels and spurious attributes.

Figure 1: An illustration of the Clever Hans effect as an analogy for spurious correlations in machine learning. Just as Clever Hans appeared to solve arithmetic problems by responding to subtle cues from his trainer, machine learning models can achieve high accuracy by exploiting spurious features, e.g., associating grass with cows rather than learning true underlying concepts. (Image generated by GPT-4o)

For example, in the training set $\mathcal{D}_{tr}$, one might observe two spurious correlations, $\langle y, a \rangle$ and $\langle y', a \rangle$, meaning that the same spurious attribute $a$ appears in two classes. A model trained on $\mathcal{D}_{tr}$ might simply use $a$ to predict $y$, which would lead to incorrect predictions on samples with the correlation $\langle y', a \rangle$. This failure to generalize across environments due to reliance on $a$ is visualized in Figure 2, where a spurious correlation between background and label (e.g., cows appearing on green grass) breaks under an environment shift (e.g., cows in the desert), while the true label remains unchanged.

### 2.1   Where do Spurious Correlations Come From?

Spurious correlations often arise from selection biases in datasets. Datasets with limited data are often underspecified, such that multiple plausible hypotheses can describe the data equally well. Following Occam's Razor, the simplest hypothesis may be preferred even if it relies on spurious patterns. This tendency reflects a more general phenomenon known as simplicity bias, where models develop an over-reliance on highly

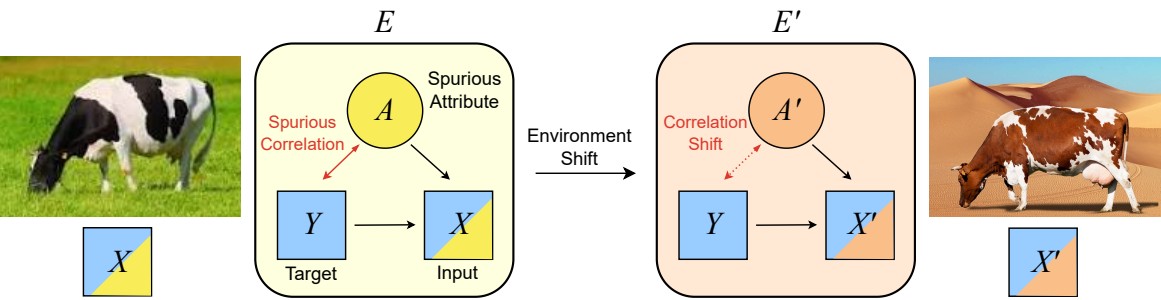

Figure 2: Depiction of a spurious correlation between spurious attribute $A$ (e.g., grass) and target $Y$ (e.g., cow). In environment $E$, input $X$ contains both core features (invariant across environments) from $Y$, and spurious features from $A$. When shifting to a new environment $E'$, the spurious attribute changes (e.g., desert background $A'$), causing the spurious correlation to break. The core features remain predictive of $Y$, but models trained on $E$ may fail to generalize if they overly rely on $A$.

available but less predictive features, as opposed to less available but highly predictive features (Hermann et al., 2023). Moreover, imbalanced group labels can lead to over-representation of certain groups, causing the model to depend on correlations that do not hold for minority groups. Finally, sampling noise (i.e., random variations inherent in collected samples) can cause the datasets to misrepresent the true data distribution, leading the model to interpret noise as a meaningful correlation.

## 2.2 Why Are Machine Learning Models Sensitive to Spurious Correlations?

**Inductive Biases from Learning Algorithms.** The "no free lunch" theorem underscores that no single model is universally optimal and that each model incorporates an inductive bias (i.e., a set of assumptions about unseen data) that influences its predictions (Wolpert & Macready, 1997). For example, CNNs are inherently biased toward local connectivity and spatial invariance, properties that are suitable for image data. However, when training data are biased, such inductive biases may lead the model to latch onto spurious patterns, resulting in overfitting and poor generalization.

**Optimization.** In the classical Empirical Risk Minimization (ERM) framework (Vapnik, 1991), we consider data grouped by group labels $g = (y, a)$, where $y$ denotes the target label and $a$ denotes a spurious attribute. Conditioning on $g$ fixes both $y$ and $a$, and the input $x$ is drawn from the conditional distribution $P(x \mid g)$. Given a model with parameters $\theta$, denoted $f_\theta : \mathcal{X} \to \mathbb{R}^K$, and a loss function $\ell : \mathbb{R}^K \times \mathcal{Y} \to \mathbb{R}$ (e.g., softmax cross-entropy), the training objective is

$$\mathcal{L}_{\text{avg}}(f_\theta) \coloneqq \mathbb{E}_{g \sim P_g} \mathbb{E}_{x \sim P(x|g)}[\ell(f_\theta(x), y)]. \tag{1}$$

Because the optimization process does not take the spurious attribute $a$ into account, if there is a strong correlation between $y$ and $a$, the model may learn to rely on $a$ when predicting $y$. In testing scenarios where the correlation does not hold, this reliance results in poor performance. It has been shown that models trained via ERM exhibit high worst-group errors, as measured by

$$\mathcal{L}_{\text{wg}}(f_\theta) \coloneqq \max_{g \in \mathcal{G}} \mathbb{E}_{x \sim P(x|g)}[\ell(f_\theta(x), y)]. \tag{2}$$

Methods such as Group Distributionally Robust Optimization (Group DRO) (Sagawa et al., 2019) have been proposed to mitigate this effect by modifying the training objective to minimize the training loss within the worst-performing groups.

## 2.3 Theoretical Insights

Recent theoretical works have expanded our understanding of how spurious correlations emerge and persist within the latent space. Analyzing the latent space of a model can reveal how core and spurious features

interact during training. Here, we use the term *core features $x_c$* to refer to the invariant predictors of the label $y$, retaining a stable, causally meaningful relationship with $y$ across environments. In contrast, $a$ represents spurious features whose association with $y$ arises from environment-specific correlations and fails to generalize under distribution shift.

The presence of spurious correlations alters not only the features that are learned but also the dynamics of learning. Recent studies consistently observe that ERM-trained models often embed spurious features early during the training process because they provide simple, highly predictive signals. Not only does this delay the learning of core features, remnants of the spurious structure are retained in the representation space even as generalizable features are eventually learned (Qiu et al., 2024). Late-stage learning can even reinforce simplified representations that encode spurious patterns (Tsoy & Konstantinov, 2024).

Bombari & Mondelli (2023) demonstrates that the degree of memorization of a spurious correlation can be geometrically characterized by the feature alignment between the spurious pattern and the training sample within the learned representation. Subsequent studies have challenged the reliability of post-hoc mitigation strategies. For instance, in two-agent rationale-predictor settings, spurious cues can originate from the rationale generator rather than the dataset (Liu et al., 2025a), while the effectiveness of regularization methods highly depends on the alignment between spurious attributes and both known and unknown concepts, as regularizers can suppress useful signals when concept correlations are entangled (Hong et al., 2025). Together, these works characterize spurious correlations as a persistent and structural phenomenon within the learned representation that indirectly results from the training objective.

## 3 Related Areas

We first provide an overview of research fields closely related to spurious correlations, offering a broader context and complementary perspectives. **Domain Generalization** is a wider goal that aims to improve model performance on unseen distributions, and it often fails when models rely on spurious correlations that do not hold across domains. **Group Robustness** focuses on ensuring consistent performance across subgroups, and spurious correlations that vary between groups can cause severe performance drops in minority groups. **Shortcut Learning** arises when models exploit spurious correlations as easy-to-learn signals instead of capturing the underlying causal features. **Simplicity Bias** leads models to favor spurious correlations because they often correspond to low-complexity patterns that are easier to optimize. We discuss each term as follows.

**Domain Generalization.** Also known as out-of-distribution (OOD) generalization, domain generalization aims to train models on one or more related domains such that they generalize well to unseen test domains (Wang et al., 2023). However, spurious correlations are a significant threat to domain generalization (Yang et al., 2022a). The core challenge in DG arises because models can easily learn features that are predictive in the source domain but become unreliable or irrelevant in a new, unseen target domain. These learned, non-causal associations are spurious correlations.

The relationship between DG and spurious correlation is *inverse* and *adversarial*. The success of any DG method is contingent on its ability to avoid or mitigate spurious correlations. For instance, in Multi-Source Domain Generalization (MSDG) (Li et al., 2018; Ahuja et al., 2021), a model might learn a feature that happens to be correlated with a label across several, but not all, possible domains. While this feature may improve performance on the available source data, it is a spurious shortcut that will likely fail in a target domain where the correlation does not exist. The problem is even more acute in Single Domain Generalization (SDG) (Li et al., 2021; Wan et al., 2022; Ilke et al., 2022), where the model has even less data diversity to learn from, making it highly susceptible to rely on spurious features specific to that one domain.

**Group Robustness.** Group robustness methods seek to ensure that a model performs consistently well across different predefined data groups or subpopulations (Yang et al., 2023c) in the data, rather than achieving high overall accuracy at the expense of minority groups (Sagawa et al., 2019). This concept is closely related to fairness and worst-case optimization: if a model latches onto a spurious feature that is prevalent in a majority group, its performance can degrade sharply on a minority group where that correlation does not hold. Spurious correlations often manifest as group-dependent performance gaps. For instance, in the Wa-

terbirds dataset (Sagawa et al., 2020), a bird classification model may learn to associate "water background" with the label waterbird (since most training images of waterbirds have water in the background), which leads to poor accuracy on the minority group of waterbirds on land. Group robustness techniques address this by actively discouraging the model from exploiting these group-specific shortcuts. One prominent approach is Group DRO, which minimizes the worst-case loss over all groups rather than the average loss (Sagawa et al., 2020). By focusing on the worst-performing group during training, the model is pushed to learn features that work for all groups, thereby reducing its reliance on spurious cues present only in the majority group. Follow-up research has proposed various improvements, such as up-weighting under-represented group examples or using two-stage training to identify and then rectify bias against minority groups (Liu et al., 2021; Idrissi et al., 2022). Notably, benchmarks such as WILDS (Koh et al., 2021) include several real-world spurious correlation tasks (e.g., Waterbirds, CelebA hair color classification, and CivilComments toxicity classification) explicitly designed to evaluate group robustness. It measures the model's performance on the worst-group (e.g., waterbirds on land, images of a certain attribute in the minority context). By treating spurious correlation issues as a group shift problem, group robustness approaches provide effective tools to diagnose and improve the model's worst-case performance under distribution shifts.

**Shortcut Learning.** Shortcut learning refers to the phenomenon where models rely on spurious correlations (often easier to learn) as opposed to the more relevant but complex features. These shortcuts are essentially spurious correlations that hold in the training set. For example, a vision model might classify images by background or texture instead of recognizing the animal itself, a classic Clever Hans behavior that breaks when the context changes. Empirical studies have shown that ImageNet-trained convolutional networks tend to prefer texture over object shape as a recognition cue, which is one type of shortcut. This bias leads to poor generalization on image distortion by shapes or image changes in domain (Geirhos et al., 2020). In NLP, a model might learn to rely on the presence of specific keywords as a shortcut on sentiment or entailment, rather than truly understanding language. It can cause errors on examples where those keywords appear in an unrelated context (Du et al., 2022a). Shortcut learning is particularly pernicious under standard ERM training because the learner will gravitate to any discriminative feature that improves training accuracy, without regard for causal relevance. Research in this area often evaluates models on "counterfactually" adjusted or stress-test datasets. For instance, testing the camel vs. cow classifier on images where camels appear on grass or cows on sand reveals the shortcut reliance. Approaches to mitigate shortcut learning overlap with those for spurious correlation in general, including data augmentation to break the spurious association, specialized training objectives, or interpretability-driven methods to detect when a model focuses on the wrong features. In summary, shortcut learning encapsulates the model's temptation to "cheat" by using easy correlations, underscoring why robust evaluation beyond the i.i.d. test set is crucial for detecting spurious reasoning.

**Simplicity Bias.** Simplicity bias (SB) is the tendency of deep neural networks (especially when trained with standard gradient-based optimization) to preferentially learn simple patterns or heuristics first, often to the exclusion of more complex but relevant features (Shah et al., 2020). In other words, given multiple predictive signals in the data, networks biased by simplicity will latch onto the easiest-to-fit signal (e.g., a low-level texture or an easily segregated background color) before considering more intricate structures (like shape combinations or abstract relations). This bias provides a fundamental reason why spurious correlations can dominate models' predictions: if a spurious feature offers a simpler decision rule that fits the training data (even if it's not fundamentally causal), the model is likely to adopt it due to SB. Shah et al. (2020) demonstrates that neural nets can indeed entirely ignore more informative features and instead rely on a less informative but simpler feature, resulting in poor generalization whenever the simple feature's correlation with the label changes. In the context of spurious correlations, SB means the model might, for example, focus on background scenery to classify objects because backgrounds are simpler to learn (roughly uniform for each class), even if object-specific features would be more reliable outside the training distribution (Tiwari & Shenoy, 2023). This predisposition is closely related to the model's vulnerability to distribution shifts and even adversarial perturbations: a network that heavily relies on a few simple features can be brittle, as small input changes can disrupt those features. Recent work has begun to address simplicity bias by altering training curricula or model architectures to force the learning of more complex features (Tiwari & Shenoy, 2023). However, SB remains a double-edged sword. It may help find a decision rule consistent with training data (an Occam's Razor effect), but it often aligns tightly with spurious correlations, thereby undermining

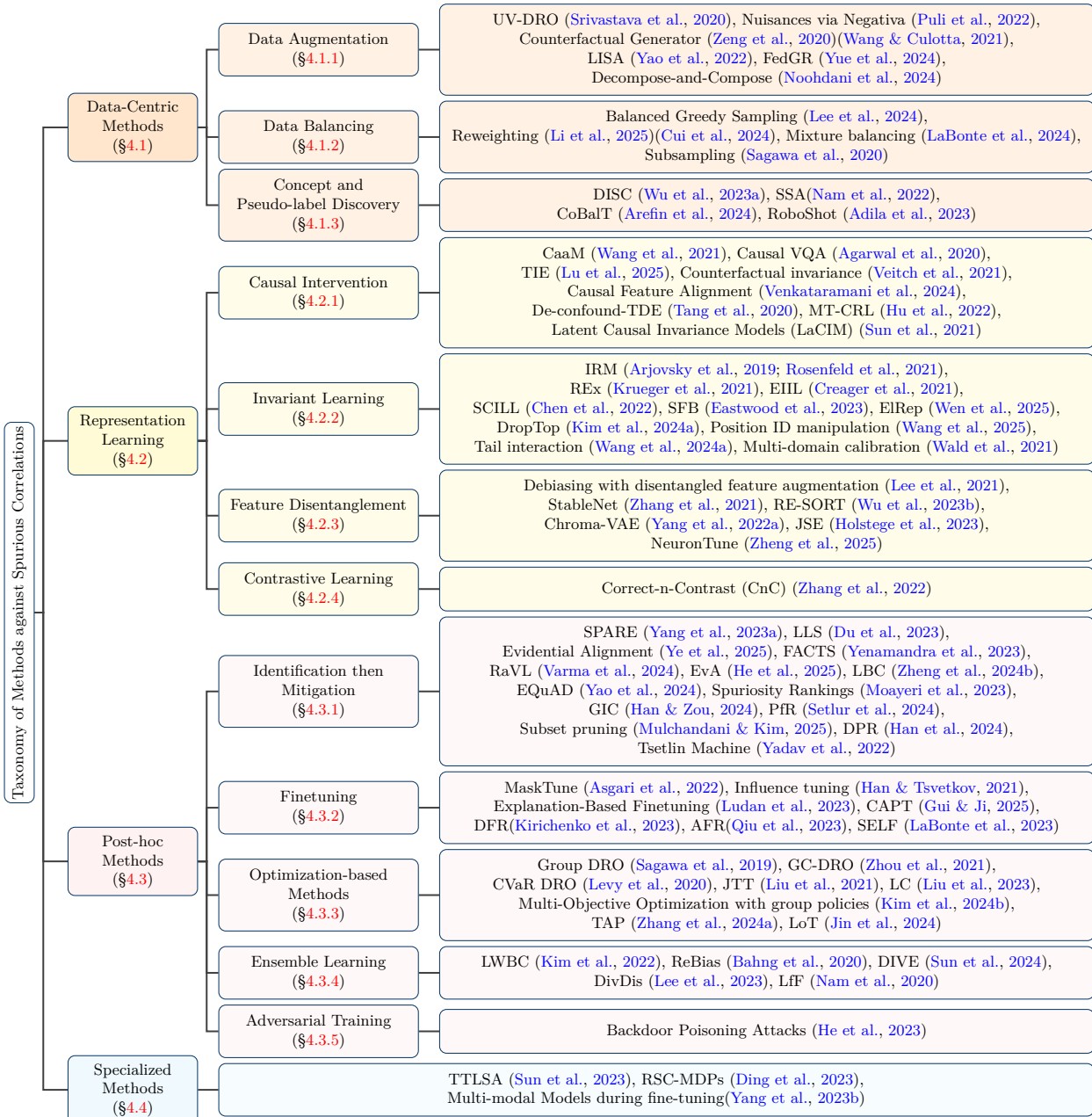

Figure 3: A comprehensive taxonomy of approaches to address spurious correlation in machine learning.

robustness. Recognizing and mitigating simplicity bias is thus important for developing models that resist spurious correlations, ensuring that they do not overly prefer "easy" but misleading patterns at the cost of true generalization.

# 4 Methods

In this section, we discuss how various machine learning techniques address the issue of spurious correlation. To provide a clear framework, we organize these methods based on their stages in the machine learning model training pipeline: **Data-Centric Methods** (Section 4.1), which modify the training data itself; **Representation Learning** (Section 4.2) approaches, which alter the model's training objective or architecture

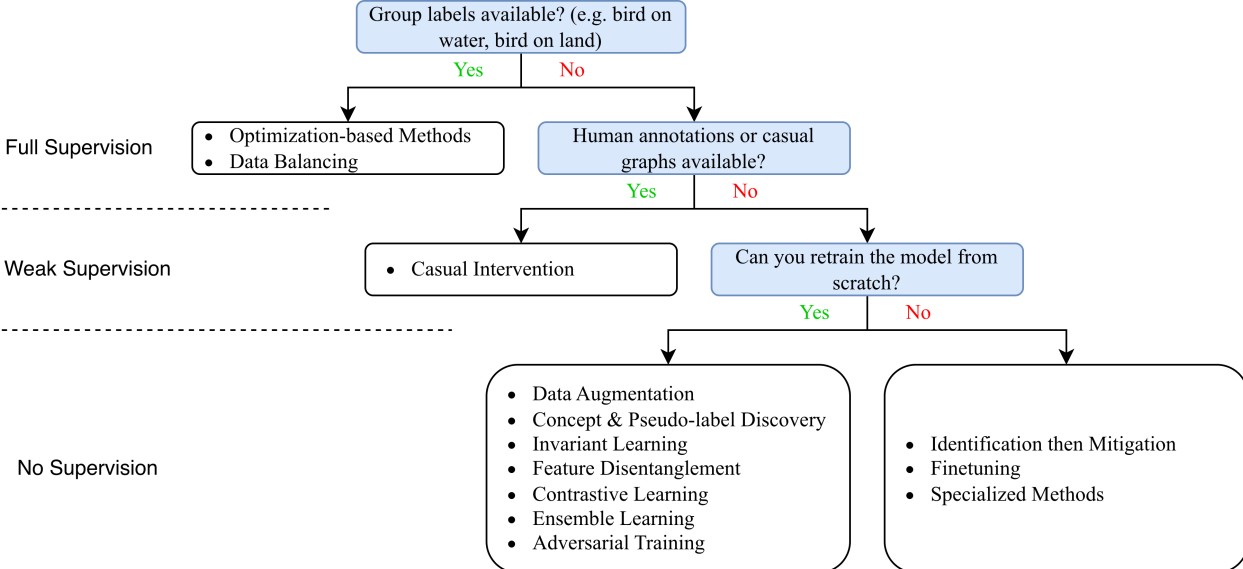

Figure 4: Operational decision map to narrow down the range of method categorization. There are some exceptions in the categories: (1) In **Optimization-based Methods**, *JTT* and *GC-DRO* do not require group labels; (2) In **Invariant Learning**, *IRM* requires group labels; (3) In **Data Augmentation**, *UV-DRO* requires human annotations; (4) In **Causal Intervention**, *CaaM* does not use annotations.

to learn more robust features; **Post-hoc Methods** (Section 4.3), which are applied to correct or refine an already trained model; and **Specialized Methods** (Section 4.4) tailored to non-standard settings. Figure 3 illustrates a comprehensive taxonomy of the surveyed approaches. Figure 4 shows a decision map for researchers to choose categories of methods based on their experiment setups, ranging from full supervision to no supervision of spurious bias.

## 4.1   Data-Centric Methods

Data-centric methods address spurious correlations at their source: the training data itself. Based on the principle that biases are learned from flawed or imbalanced data distributions, these techniques aim to modify the input data before or during training to break the harmful associations between spurious attributes and labels. This category encompasses a range of strategies, from **Data Augmentation** (Section 4.1.1) techniques that create new examples to expose the model to more diverse contexts, to **Data Balancing** (Section 4.1.2) methods that correct for distributional imbalances, such as reweighting and subsampling. Other approaches focus on **Concept and Pseudo-label Discovery** (Section 4.1.3) to explicitly identify and manage spurious attributes using labeled or unlabeled data.

### 4.1.1   Data Augmentation

Data augmentation increases the diversity of a dataset without the need for new data collection. Techniques such as image rotation, cropping, and noise injection are used to produce varied training samples, thereby improving model generalization by mitigating the impact of spurious correlations. For example, UV-DRO (Srivastava et al., 2020) leverages human annotations to augment training examples with potential unmeasured variables, reducing the spurious correlation problem to a covariate shift problem. When additional annotations are unavailable, methods such as Nuisances via Negativa (Puli et al., 2022) corrupt semantic information to highlight nuisances, while LISA (Yao et al., 2022) employs a mixup-based technique for selective augmentation. In graph settings, FedGR (Yue et al., 2024) introduces anti-shortcut augmentations by partitioning the graph into rationale and non-rationale subgraphs, perturbing the latter to dissociate spurious cues from the target signal, while also leveraging differences between local and global models within a federated learning paradigm.

Another class of augmentation approaches relies on generating counterfactual examples, or modified training samples that isolate causal and non-causal factors. This trains the model to ignore irrelevant details by separating spurious and true signals. The Counterfactual Generator (Zeng et al., 2020) creates counterfactual examples by intervening on existing observations, while Wang & Culotta (2021) proposes training a robust text classifier by automatically generating counterfactual training data. Decompose-and-Compose (Noohdani et al., 2024) takes an alternative approach, deconstructing images into causal and non-causal segments before recombining them into new samples.

### 4.1.2  Data Balancing

Spurious correlations are often caused by distributional imbalances in the training data, where spurious features consistently co-occur with target labels despite having no causal relationship. These issues can be addressed across various domains by modifying how existing data is sampled or weighted during training. For example, Balanced Greedy Sampling (Lee et al., 2024) is found to prevent such biases in continual learning tasks by retraining the final layer of the model using a balanced training subset. Reweighting has been used to mitigate dataset bias resulting from dataset condensation (Cui et al., 2024) and shortcut bias in multimodal settings (Li et al., 2025). However, LaBonte et al. (2024) warns that common group-balancing techniques can fail in specific scenarios: mini-batch upsampling and loss upweighting can observe a decrease in worst-group accuracy in later training epochs, and the effectiveness of data pruning varies depending on the group structure. Instead, they find that a combination of the two methods achieves a higher worst-group accuracy than each method individually. Similarly, Sagawa et al. (2020) found that on overparameterized models, the subsampling on the majority group of data brings lower worst-group error instead of upweighting the minority group.

### 4.1.3  Concept and Pseudo-label Discovery

Predefining concepts and generating pseudo-labels to enhance the conceptual structure of the training data can help to break intrinsic spurious correlations and improve model robustness. For instance, DISC (Wu et al., 2023a) discovers unstable predefined concepts across different environments, which are then used to augment the training data. Similarly, SSA (Nam et al., 2022) generates pseudo-labels for spurious attributes using a limited amount of annotated data, and the pseudo-labeled attributes are then combined with the training data to train a robust model using worst-case loss minimization. Recent methods have explored unsupervised or zero-shot discovery of latent concepts to avoid the need for human labeling. CoBalT (Arefin et al., 2024) uses various clustering techniques to identify concepts, while RoboShot (Adila et al., 2023) uses a language model to generate concept subspaces by extracting the embedding of the task description from within a language model.

### 4.1.4  Discussion

Data-centric methods are greatly beneficial because they operate at the source of the spurious correlations and can often be combined with standard ERM training. However, there exists significant limitations to this approach. Augmentation and balancing strategies assume that spurious attributes are at least partially observable or manipulable. In cases where spurious cues are subtle, high-dimensional, or entangled with core features, these methods can fail to break shortcut patterns or inadvertently distort the underlying causal signal. Moreover, recent work (LaBonte et al., 2024; Sagawa et al., 2020) has shown that naive group balancing or sub-sampling can *worsen* worst-group performance in later training stages, and that the effectiveness of reweighting and pruning is highly sensitive to the true group structure and model capacity. As a result, data-centric approaches offer strong gains in settings with reasonably well-understood spurious structure, but can be brittle or even harmful when the assumed bias model is mis-specified.

### 4.2  Representation Learning

In contrast to data-centric approaches, representation learning methods shift the focus from the data itself to the model's internal learning process and the quality of its learned features. By modifying the training objective or model architecture, these approaches allow the model to identify features that are inherently

robust and causally linked to the outcome, rather than just statistically correlated. Key strategies include **Causal Intervention** (Section 4.2.1), which explicitly models the data's causal structure, and **Invariant Learning** (Section 4.2.2), which seeks features that remain stable across diverse environments. Other techniques focus on altering the latent space directly, such as **Feature Disentanglement** (Section 4.2.3), which aims to separate spurious and core features, and **Contrastive Learning** (Section 4.2.4), which enforces representational similarity for inputs that differ only in their spurious attributes.

### 4.2.1 Causal Intervention

Causal intervention methods focus on explicitly addressing the causal relationships between the input, label, and potential spurious attributes, aiming to improve model robustness and fairness by reducing the influence of spurious features. For example, CaaM (Wang et al., 2021) integrates a Causal Attention Module within the Vision Transformer architecture, using unsupervised learning to self-annotate and mitigate confounding effects. Similarly, Causal VQA (Agarwal et al., 2020) applies a semantic editing-based approach to assess and improve the robustness of Visual Question Answering (VQA) models, and TIE (Lu et al., 2025) leverages a translation operation within image embeddings based on text embeddings along a given spurious vector. Venkataramani et al. (2024) proposes Causal Feature Alignment, which uses a trained ERM classifier to extract core features within an image. Veitch et al. (2021) and Sun et al. (2021) incorporate structured causal graphs to reflect true relationships in the data, while De-confound-TDE (Tang et al., 2020) employs causal modeling to pinpoint the effect of momentum in long-tailed classification tasks. In multi-task settings, Multi-Task Causal Representation Learning (MT-CRL) (Hu et al., 2022) identifies the causal structure between different tasks to regularize the learning process, preventing the model from relying on spurious features that are shared across non-causally related tasks. In a similar vein, Veitch et al. (2021) introduces counterfactual invariance to evaluate the causal structure captured within the model by measuring whether predictions are influenced when changing irrelevant portions of a sample.

### 4.2.2 Invariant Learning

Invariant learning methods aim to train models to identify and focus on features that remain stable across different training environments, based upon the assumption that truly predictive features are invariant to environmental shifts. This strategy is supported by Wald et al. (2021), which establishes that models achieving multi-domain calibration are provably free of spurious correlations. Early work such as IRM (Arjovsky et al., 2019) formulates an objective to learn deep invariant features that capture complex relationships between latent variables. Methods like REx (Krueger et al., 2021) and EIIL (Creager et al., 2021) further this goal by focusing on risk-level consistency and automatic environment inference, respectively, and SCILL (Chen et al., 2022) establishes theoretical group criteria to ensure group-invariant learning. Recent works emphasize enforcing invariance at the feature level. SFB (Eastwood et al., 2023) selectively amplifies features that are stable across domains. ElRep (Wen et al., 2025) introduces nuclear and Frobenius norm penalties on the representation matrix to balance sparsity and smoothness, improving generalization while avoiding over-regularization. DropTop (Kim et al., 2024a) dynamically identifies and debiases shortcut features in continual learning by selecting top-k activations that correlate with spurious behavior. In the Large Language Model (LLM) setting, Wang et al. (2025) demonstrates how manipulating position encodings can eliminate hidden shortcuts in role separation tasks, reinforcing invariant structure. Wang et al. (2024a) disentangles spurious and beneficial correlations by combining frequency-restriction and interaction-based modules, enhancing the model's ability to learn domain-invariant representations.

### 4.2.3 Feature Disentanglement

Feature disentanglement methods aim to separate spurious representations from general representations in the latent space. These approaches often use dual-branch architectures. For example, StableNet (Zhang et al., 2021) removes both linear and non-linear dependencies using random Fourier features combined with a standard classifier, while Lee et al. (2021) trains two encoders to extract bias and intrinsic features separately. Disentanglement can be further enhanced through high- or low-dimensional projections. RE-SORT (Wu et al., 2023b) utilizes a Laplacian kernel function to project feature interactions to a higher dimension from within a multilevel stacked recurrent structure before targeted elimination through sample reweighting.

In contrast, Chroma-VAE (Yang et al., 2022a) employs a dual-pronged VAE to disentangle latent representations in a low-dimensional subspace, ensuring that shortcut features do not dominate the learning process, while Holstege et al. (2023) jointly identifies two low-dimensional orthogonal subspaces within a vector representation to locate encoded spurious features prior to removal (JSE). Similarly, NeuronTune (Zheng et al., 2025) identifies the neurons in latent space that are affected by spurious correlations, and zeros out these neurons during the retraining process.

### 4.2.4 Contrastive Learning

Contrastive learning, a popular approach within self-supervised learning, has shown promise in addressing spurious correlations. The Correct-n-Contrast (CnC) method (Zhang et al., 2022) first trains an ERM model to identify samples within the same class that vary in spurious features, and then uses contrastive learning to encourage similar representations for these samples. The resulting model is better able to distinguish between essential and non-essential features.

### 4.2.5 Discussion

Representation learning approaches directly tune latent features so that core information is preserved and spurious information is suppressed. In practice, however, they face several trade-offs. Causal-intervention methods often require a partial causal graph, interventional data, or strong modeling assumptions, and can degrade in-distribution accuracy if the assumed causal structure is incomplete or incorrect. When environments do not sufficiently vary spurious factors, invariant learning methods such as IRM and its variants can either have little effect or over-regularize useful correlations. Disentanglement and contrastive techniques rely on the availability of pairs or groups of examples that isolate specific factors of variation, and identifiability issues can cause seemingly disentangled units to still mix core and spurious signals. Thus, while representation-level methods can provide significant theoretical guarantees in ideal settings, their empirical success hinges on how well their structural assumptions match the underlying data-generating process.

## 4.3 Post-hoc Methods

Post-hoc methods are designed to mitigate spurious correlations in models that have already been trained, or during a secondary training stage like fine-tuning. Rather than building a robust model from scratch, these techniques typically operate on a pre-existing, potentially biased model, aiming to diagnose and correct its reliance on spurious features without full retraining. This diverse category includes **Identification then Mitigation** (Section 4.3.1) strategies that first find bias-conflicting samples and then reduce their influence; **Finetuning** (Section 4.3.2) approaches that refine a pre-trained model on carefully selected or transformed data; and **Optimization-based Methods** (Section 4.3.3) like Group DRO that modify the training objective to improve worst-group performance. Other prominent techniques involve **Ensemble Learning** (Section 4.3.4) to combine multiple biased models for a more robust prediction, and **Adversarial Training** (Section 4.3.5) to improve model resilience.

### 4.3.1 Identification then Mitigation

A number of methods follow an "identification then mitigation" strategy. These approaches first identify samples that are likely affected by spurious correlations and then mitigate their impact. SPARE (Yang et al., 2023a) and LLS (Du et al., 2023) both identify spurious correlations based on signs of simplicity bias, and use importance sampling or reweighting to reduce bias. Evidential Alignment (Ye et al., 2025) utilizes second-order risk minimization to identify and quantify the spurious correlations in pretrained models, then tunes the biased model based on overconfident samples from a calibration dataset. Spurious correlations can also be identified in the model's representations through latent feature probing. FACTS (Yenamandra et al., 2023) amplifies correlations to fit a bias-aligned hypothesis and then uses slicing via mixture modeling in the corresponding feature space to address under-performing data slices. RaVL (Varma et al., 2024) creates a region-aware loss function based on local image features, and EvA (He et al., 2025) learns class-specific spurious indicators within the model. LBC (Zheng et al., 2024b) and EQuAD (Yao et al., 2024) both quantify the likelihood and strength of spurious features through projection into a spurious embedding

space and into a low-dimensional latent space, respectively. Mulchandani & Kim (2025) finds that spurious correlations result from a small subset of the data and introduces a novel pruning technique that identifies and removes such samples. Disagreement Probability-based Resampling (DPR) (Han et al., 2024) takes the reverse approach, identifying and upsampling training examples that do not align with spurious correlations. In addition, Yadav et al. (2022) transforms natural language data into rule-based logic formulations and employs logical negation via a Tsetlin Machine to achieve explainable debiasing. Other methods like GIC, Spuriosity Rankings, and PfR infer group membership using auxiliary models or heuristic signals. GIC trains a classifier to predict group labels from spurious-label correlations, PfR uses zero-shot vision-language model predictions to identify spurious features, and Spuriosity Rankings proxies spuriosity using interpretable neural features to rank examples within each class and fine-tune on less biased data (Han & Zou, 2024; Moayeri et al., 2023; Setlur et al., 2024).

### 4.3.2 Finetuning

Finetuning strategies focus on refining a pre-trained general model by selectively adjusting sections of either the model or the dataset to reduce reliance on spurious features. For instance, MaskTune (Asgari et al., 2022) encourages the model to consider a wider array of input variables by mapping them to the same target, while Influence Tuning (Han & Tsvetkov, 2021) backpropagates attribution information for targeted refinement. In the text domain, Explanation-Based Finetuning (Ludan et al., 2023) trains models on artificially constructed datasets containing spurious cues, then tests on clean sets. Similarly, Causality-Aware Post-Training (CAPT) (Gui & Ji, 2025) fine-tunes models on a transformed dataset where specific events are first identified and then replaced with randomized, abstract symbols to break spurious correlations acquired during pre-training. Moreover, last-layer finetuning strategies such as Deep Feature Reweighting (DFR) (Kirichenko et al., 2023; Izmailov et al., 2022), Automatic Feature Reweighting (AFR) (Qiu et al., 2023), and Selective Last-Layer Finetuning (SELF) (LaBonte et al., 2023) have been shown to improve worst-group performance with minimal group annotations. Shuieh et al. (2025) compares various fine-tuning techniques, finding Supervised Fine-tuning to excel in complex, context-intensive tasks, while Direct Preference and Kahneman-Tversky Optimization both perform well in mathematical reasoning tasks.

### 4.3.3 Optimization-based Methods

Optimization-based strategies modify the training objective to promote better performance under spurious correlations. Group DRO (Sagawa et al., 2019) minimizes the worst-case loss across different groups, ensuring robust performance across subpopulations. GC-DRO (Zhou et al., 2021) extends group DRO by jointly reweighting groups and individual instances (without needing perfectly clean group partitions or a noise model), thereby overcoming group DRO's failure under imperfect or mismatched groupings. CVaR DRO (Levy et al., 2020) extends this idea by optimizing the expected loss over the worst-case data distribution. Just Train Twice (JTT) (Liu et al., 2021) identifies and upsamples informative training samples with high training loss. Additionally, Logit Correction (LC) (Liu et al., 2023) adjusts model output logits based on group membership to balance performance across groups. In a similar vein, Kim et al. (2024b) introduces a Multi-Objective Optimization function that minimizes group-wise losses before adjusting weights to resolve conflicting objectives. Within deep CNNs, TAP (Zhang et al., 2024a) prevents spurious correlation reemergence by penalizing activations in later layers. LoT (Jin et al., 2024) provides a unique knowledge distillation training loop, where a teacher model trains student models to provide feedback to the main model, helping it capture invariant generalizations.

### 4.3.4 Ensemble Learning

Ensemble learning methods combine multiple independently trained biased models to produce an overall debiased prediction. For instance, LWBC (Kim et al., 2022) uses a committee of classifiers to identify bias-conflicting data and then emphasizes these samples during the main classifier's training. Similarly, ReBias (Bahng et al., 2020) trains de-biased representations by enforcing statistical independence from intentionally biased representations. DIVE (Sun et al., 2024) trains a collection of models on diverging subgraphs to encourage learning distinct patterns from the overall predictive graph. Some methods, such as DivDis (Lee

et al., 2023) and LfF (Nam et al., 2020), adopt a two-stage ensemble strategy to identify and mitigate reliance on spurious features.

### 4.3.5 Adversarial Training

Adversarial training methods strengthen the model against adversarial inputs, thereby reducing reliance on spurious correlations. For instance, methods addressing backdoor poisoning attacks consider the spurious correlations introduced by watermarked, mislabeled training examples and propose mitigation strategies based on adversarial training (He et al., 2023).

### 4.3.6 Discussion

Post-hoc methods can reuse existing high-performing models and avoid retraining from scratch, but this convenience comes with distinct risks. Identification-then-mitigation approaches rely on proxy signals (loss, disagreement, feature attributions, auxiliary classifiers) to detect spurious reliance. These signals are often imperfect: high loss may reflect intrinsic difficulty rather than bias; disagreement can arise from model underfitting rather than shortcut learning; and attribution maps are known to suffer from attribution bias and instability, sometimes highlighting spurious regions even when the model is not relying on them. Fine-tuning and optimization-based strategies such as Group DRO and its variants often require group labels or reasonably accurate surrogates. Noisy or incomplete group information causes reweighting strategies to overfit small slices or harm majority performance without improving robustness. Ensemble and adversarial schemes increase computational cost and can degrade overall performance, especially when the adversarial or bias models are poorly aligned with real-world shifts. Overall, post-hoc methods provide flexible tools for patching existing systems, but their effectiveness is tightly coupled to the quality of their bias-identification signals and the stability of secondary training.

## 4.4 Specialized Methods

This category includes approaches designed for specific settings. Test-Time Label-Shift Adaptation (TTLSA) (Sun et al., 2023) addresses label shifts at test time, and RSC-MDP (Ding et al., 2023) tackles spurious dependencies in reinforcement learning contexts. Additional work (Yang et al., 2023b) explores the use of multi-modal models to explicitly separate spurious attributes from the main class. Specialized methods highlight that spurious correlations manifest differently across domains. These techniques achieve strong performance within their target setting by leveraging domain-specific structures, but their robustness benefits do not always transfer to more general supervised learning scenarios. Nonetheless, they surface important domain-dependent failure modes which can inform the design of more broadly applicable debiasing strategies.

# 5 Datasets and Metrics

## 5.1 Datasets

This subsection provides an overview of benchmark datasets used for studying spurious correlations in machine learning, as summarized in Table 1. In these datasets, the correlation between the label $y$ and a spurious attribute $a$ observed during training is not guaranteed to hold during testing. The reviewed benchmarks, which are publicly available via the hyperlinks in Table 1, are categorized into five domains: Vision, Text, Graph, Multimodal, and Health. These datasets are constructed using both realistic and synthetic data, with synthetic generation being an increasingly prevalent approach (Qian et al., 2023). The distinct methodologies used to create both synthetic and realistic datasets are discussed below.

### 5.1.1 Synthetic Datasets

Synthetic datasets are created to enable a more controllable analysis of spurious correlations in machine learning models. Since spurious features do not appear systematically or easily isolated in standard benchmarks, manually synthesizing spurious features improves the efficiency of creating relevant datasets. Different types of spurious attributes could be manipulated to create challenging datasets targeting the model's robustness.

Table 1: Datasets used to study spurious correlations in machine learning. Datasets are grouped by domain. The Synthetic column indicates whether a dataset is synthetic (✓) or derived from natural data (✗).

| Dataset | Source Paper | #Class | #Attributes | #Sample | Brief Description | Synthetic | Domain |
|---|---|---|---|---|---|---|---|
| UrbanCars | Li et al. (2023) | 2 | 2 | 8,000 | Compositional vehicle scenes | ✓ | Vision |
| Colored MNIST | Arjovsky et al. (2019) | – | 2 | 60,000 | Color-injected digits | ✓ | Vision |
| CIFAR-10-C | Hendrycks & Dieterich (2019) | – | 15 | 60,000 | Corruption on CIFAR-10 | ✓ | Vision |
| SpuCo | Joshi et al. (2023) | 4 | 4 | 118,100 | Mixed synthetic + natural data | ✓ | Vision |
| Waterbirds | Sagawa et al. (2019) | 2 | 2 | 11,788 | Composite birds + backgrounds | ✓ | Vision |
| ImageNet-C | Hendrycks & Dieterich (2019) | 75 | – | – | Corruption benchmark (Variant of ImageNet) | ✓ | Vision |
| ImageNet-R | Hendrycks et al. (2021a) | 200 | 16 | 30,000 | Artistic renditions (Variant of ImageNet) | ✓ | Vision |
| ImageNet-W | Li et al. (2023) | - | 2 | - | ImageNet with Watermark (Variant of ImageNet) | ✓ | Vision |
| ImageNet-9 | Xiao et al. (2020) | 9 | – | 5,495 | Background-altered (Variant of ImageNet) | ✓ | Vision |
| Stylized-ImageNet | Geirhos et al. (2019) | – | – | – | AdaIN stylised variant (Variant of ImageNet) | ✓ | Vision |
| Hard ImageNet | Moayeri et al. (2022) | 15 | – | 19,847 | Spurious correlation specific (Subset of ImageNet) | ✗ | Vision |
| ImageNet-A | Hendrycks et al. (2021b) | 200 | – | 7,500 | Natural adversarial images (Subset of ImageNet) | ✗ | Vision |
| PACS | Li et al. (2017) | 7 | 4 | 9,991 | Multi-domain real images | ✗ | Vision |
| VLCS | Torralba & Efros (2011) | 5 | 4 | 10,729 | Aggregated real domains | ✗ | Vision |
| Office-Home | Venkateswara et al. (2017) | 65 | 4 | 15,500 | Multi-domain real images | ✗ | Vision |
| CelebA | Liu et al. (2015) | 2 | 2 | 202,599 | Face attributes | ✗ | Vision |
| MetaCoCo | Zhang et al. (2024b) | 100 | 155 | 175,637 | Few-shot data for spurious correlations | ✗ | Vision |
| BAR | Nam et al. (2020) | 6 | 6 | 2,595 | Biased action recognition | ✗ | Vision |
| NICO | He et al. (2021) | 19 | 188 | 24,214 | Context-shift animals | ✗ | Vision |
| MetaShift | Liang & Zou (2022) | 410 | – | 12,868 | Dataset-of-datasets shift | ✗ | Vision |
| FMOW | Christie et al. (2018) | 63 | – | 1,047,691 | Satellite imagery | ✗ | Vision |
| bFFHQ | Lee et al. (2021) | 2 | 2 | – | Balanced FFHQ faces | ✗ | Vision |
| CounterAnimal | Wang et al. (2024b) | 45 | 14 | 13,100 | Counterfactual animals | ✗ | Vision |
| WILDS-iWildCam | Koh et al. (2021) | 182 | – | 203,029 | Wildlife camera traps | ✗ | Vision |
| WILDS-GlobalWheat | Koh et al. (2021) | 1 | 12 | 6,515 | Wheat images | ✗ | Vision |
| WILDS-PovertyMap | Koh et al. (2021) | - | 23 × 2 | 9,669 | Satellite images | ✗ | Vision |
| QuAC | Choi et al. (2018) | – | – | 98,407 | Conversational QA | ✗ | Text |
| POVID | Zhou et al. (2024a) | – | – | 17,000 | VLM prompt–response | ✗ | Text |
| CivilComments | Borkan et al. (2019) | 2 | 24 | 1,999,514 | Online comments | ✗ | Text |
| MultiNLI | Williams et al. (2017) | 3 | 2 | 206,175 | Natural language inference | ✗ | Text |
| FDCL18 | Founta et al. (2018) | 7 | – | 80,000 | Offensive speech | ✗ | Text |
| WILDS-Amazon | Koh et al. (2021) | - | 2 | 539,502 | Customer reviews on Amazon | ✗ | Text |
| ShortcutSuite | Yuan et al. (2024) | - | - | - | Shortcut in LLM | ✗ | Text |
| Spurious-Motif | Wu et al. (2022) | 3 | 3 | 18,000 | Compositional motif-based graph | ✓ | Graph |
| GOOD-Motif | Gui et al. (2022) | 5 | 3 | 30,000 | Compositional motif-based graph | ✓ | Graph |
| GOOD-HIV | Gui et al. (2022) | 2 | 2 | 41,127 | Molecule dataset | ✗ | Graph |
| GOOD-SST2 | Gui et al. (2022) | 2 | - | 70,042 | Sentence polarity | ✗ | Graph |
| DrugOOD | Ji et al. (2023) | – | 5 | 58,000 | Drug discovery | ✗ | Graph |
| IV/CV-VQA | Agarwal et al. (2020) | – | – | 716,603 | Causal VQA | ✗ | Multimodal |
| MM-SpuBench | Ye et al. (2024) | – | – | 2,400 | Multimodal spurious correlations | ✗ | Multimodal |
| ChestX-ray14 | Wang et al. (2017) | 14 | – | 112,120 | Chest X-ray | ✗ | Health |
| MIMIC-CXR | Johnson et al. (2019) | 14 | – | 377,110 | Chest X-ray | ✗ | Health |
| CheXpert | Irvin et al. (2019) | 14 | – | 224,316 | Chest X-ray | ✗ | Health |
| PadChest | Bustos et al. (2020) | 19 | – | 160,868 | Chest X-ray | ✗ | Health |
| COVID-19 | Cohen et al. (2020) | 2 | – | 21,165 | Chest X-ray | ✗ | Health |

In this subsection, common methodologies used to create synthetic datasets are categorized into four primary techniques: data composition, feature and noise injection, and style transfer.

**Data Composition**  A primary method for synthetic dataset creation is the composition of data from multiple sources. For instance, to analyze the spurious correlation between a foreground object and its background, images from different datasets can be combined. An example of this is the UrbanCars dataset (Li et al., 2023), where the authors integrated foreground vehicle images from the StanfordCars dataset (Krause et al., 2013) with background scenes from the Places dataset (Zhou et al., 2017). Another common practice involves placing different object types onto a shared background. This technique is used to analyze potential spurious correlations between co-occurring objects or to study shortcut learning, as demonstrated in the ImageNet-W dataset (Li et al., 2023).

**Pixel-Level Manipulation**  Manipulating existing datasets at the pixel level could also introduce desired spurious cues. While the target features still remain, the model might get biased towards the manipulation pattern. A notable example is the Colored MNIST dataset (Arjovsky et al., 2019). In this work, the authors applied color filters to the original MNIST images to test whether a model learns the digit itself or relies on the color as a shortcut. In addition, noise injection is also a common method to introduce spurious attributes. In CIFAR-10-C and ImageNet-C, Hendrycks & Dieterich (2019) applied perturbations such as noise, blurring, weather filters, and image manipulations (e.g., brightness, contrast, etc.) to the CIFAR-10 and ImageNet datasets to evaluate the robustness of models. Similarly, in the SpuCoMNIST dataset

(Joshi et al., 2023), varying levels of noise are injected into MNIST images to establish different spurious correlations.

**Style Transfer**  Machine learning models can exhibit biases toward object texture or style rather than shape. Style transfer is used to create datasets that test for this phenomenon. For example, the Stylized-ImageNet dataset (Geirhos et al., 2019) was generated by applying various artistic styles to images from ImageNet. The style transfer was performed using the AdaIN algorithm, which was trained on images from Kaggle's "Painter by Numbers" dataset (Huang & Belongie, 2017). The resulting dataset is instrumental in identifying whether a model exhibits a "shape bias" or a "texture bias".

### 5.1.2 Realistic Datasets

In contrast to synthetic datasets, which are engineered for controlled analysis, realistic datasets are curated from real-world sources to capture the natural, often subtle and unexpected, spurious correlations that models are likely to encounter in deployment. While they may not offer the same level of granular control as their synthetic counterparts, these benchmarks are indispensable for evaluating a model's performance on authentic data distributions. The curation of such datasets involves diverse methodologies, which we discuss below based on their primary data sourcing strategy.

**Data Curation from Existing Sources**  A common strategy for creating new benchmarks is to collect and fuse data from existing sources, such as public datasets or search engines. This approach is prevalent in domain generalization, where datasets are constructed by combining data from different domains. For instance, the PACS dataset (Li et al., 2017) aggregates images of the same object classes from four distinct domains, ranging from photorealistic sources such as Caltech256 (Griffin et al., 2022) to sketch-based sources such as TU-Berlin (Eitz et al., 2012). In other cases, datasets are curated by sub-sampling or re-annotating a single source. An example is bFFHQ (Lee et al., 2021), which was created by adding new annotations for a protected attribute to the original FFHQ dataset (Karras et al., 2019).

**Collection from Scratch**  Beyond leveraging existing data, some benchmarks are constructed by collecting and annotating data directly from real-world environments. This approach is critical for capturing authentic data distributions and task-specific nuances that may be absent in pre-existing datasets. This methodology is applied in various fields: the FMOW dataset (Christie et al., 2018) contains satellite images for land use classification; the CivilComments dataset (Borkan et al., 2019) comprises user-generated comments for toxicity detection; and the PadChest dataset (Bustos et al., 2020) includes chest x-ray images and corresponding radiological reports from a hospital. A defining characteristic of these datasets is the significant manual annotation effort they require, which is essential for establishing ground-truth labels.

**Mining Challenging Example**  A specialized curation method involves mining a large benchmark for examples that are inherently challenging for standard machine learning models. Unlike methods that select data based on class or domain, this technique specifically retrieves instances that are likely to elicit incorrect predictions due to strong spurious cues or other adversarial properties. For example, the Hard ImageNet dataset (Moayeri et al., 2022) consists of images from ImageNet (Russakovsky et al., 2015) that were identified as containing potent spurious features. Similarly, the ImageNet-A dataset (Hendrycks et al., 2021b) is composed of naturally occurring adversarial examples from ImageNet that consistently fool well-established classification models.

## 5.2 Metrics

A commonly used metric for quantifying robustness to spurious correlations is the **worst-group accuracy** $\text{Acc}_{\text{wg}}(f_\theta)$ (Idrissi et al., 2022; Chaudhuri et al., 2023), defined as the minimum accuracy across group-labeled test sets:

$$\text{Acc}_{\text{wg}}(f_\theta) \coloneqq \min_{g \in \mathcal{G}} \mathbb{E}_{(x,y) \sim \mathcal{D}_{\text{test}}^g} \big[ \mathbb{1}\big( f_\theta(x) = y \big) \big], \tag{3}$$

where $g = (y, a)$ is a group label determined by the target $y$ and the spurious attribute $a$.

Table 2: A meta-analysis of worst-group accuracy (WGA) and average accuracy (Avg) across four common spurious correlation benchmarks. Data is sourced from the comprehensive benchmarking by Yang et al. (2023c) under the setting where spurious attributes are unknown during training and validation. The format is presented as WGA / Avg.

| Method | Waterbirds (Vision) | CelebA (Vision) | CivilComments (Text) | MultiNLI (Text) |
|---|---|---|---|---|
| ERM | 69.1 / 84.1 | 57.6 / 95.0 | 63.2 / 85.4 | 66.4 / 81.0 |
| Group DRO | 73.1 / 86.9 | 68.3 / 94.4 | 61.5 / 81.2 | 64.1 / 81.1 |
| CVaR DRO | 75.5 / 89.9 | 60.2 / 95.1 | 62.9 / 81.6 | 48.2 / 75.4 |
| JTT | 71.2 / 88.9 | 48.3 / 95.9 | 51.0 / 79.0 | 65.1 / 81.4 |
| LfF | 75.0 / 86.6 | 53.0 / 81.1 | 42.2 / 69.1 | 57.3 / 71.4 |
| LISA | 77.5 / 89.2 | 57.8 / 95.4 | 65.8 / 84.6 | 66.8 / 81.7 |
| DFR | 89.0 / 92.2 | 73.7 / 93.6 | 64.4 / 80.7 | 63.8 / 80.2 |

Other relevant metrics include:

**Average-group accuracy.** This measures the mean accuracy across all groups:

$$\text{Acc}_{\text{ag}}(f_\theta) \coloneqq \frac{1}{|\mathcal{G}|} \sum_{g \in \mathcal{G}} \mathbb{E}_{(x,y) \sim \mathcal{D}_{\text{test}}^g} \big[ \mathbb{1}\big(f_\theta(x) = y\big) \big]. \tag{4}$$

Unlike worst-group accuracy, this metric is less sensitive to the hardest subgroup and instead reflects whether performance is balanced across groups on average.

**Bias-conflicting accuracy.** This is defined on a subset of groups $\mathcal{G}_{bc} \subseteq \mathcal{G}$ where the spurious attribute $a$ does not align with the majority correlation observed during training:

$$\text{Acc}_{\text{bc}}(f_\theta) \coloneqq \frac{1}{|\mathcal{G}_{bc}|} \sum_{g \in \mathcal{G}_{bc}} \mathbb{E}_{(x,y) \sim \mathcal{D}_{\text{test}}^g} \big[ \mathbb{1}\big(f_\theta(x) = y\big) \big]. \tag{5}$$

This metric isolates model performance on the challenging bias-conflicting cases, where reliance on spurious features is most detrimental.

Finally, regular average accuracy across the entire test set remains useful for capturing overall utility, though it does not directly reflect robustness to spurious correlations. Together, these complementary metrics provide a more holistic view of a model's dependence on spurious features. A lightweight table of methods performance on common datasets are shown in Table 2.

## 6 Broader Impacts of Spurious Correlations

### 6.1 LLM Alignment

Recent advancements in LLMs have significantly expanded their capabilities and revolutionized agentic AI assistants for challenging tasks, such as mathematical reasoning, tool use, code generation, and interactive communication. Compared to traditional machine learning settings, the length of tasks that current AI models can solve has increased exponentially from several tokens (e.g., classification) to millions of tokens (e.g., long Chain-of-Thought (CoT) reasoning and agentic workflows). However, spurious correlations in traditional machine learning still persist. For instance, image classification models have been found to use correlations between textures and image classes (Geirhos et al., 2019) for object recognition instead of focusing on defining features of objects. To solve this problem, many approaches (Kirichenko et al., 2023; Zheng et al., 2024b;c) and benchmarks (Sagawa et al., 2020; Zheng et al., 2024a) have been proposed to mitigate the spurious biases in the classification tasks.

AI assistants are typically trained to generate outputs that align with human ratings/preferences using methods such as *reinforcement learning from human feedback* (RLHF) or *preference optimization* (PO).

These techniques depend on explicit or implicit reward modeling to guide LLMs to learn human preferences in a helpful, harmless, and honest (HHH) manner (Askell et al., 2021; Bai et al., 2022). However, the spurious correlations remain a significant issue and may introduce severe threats. It has been evident that the learned reward within the AI assistants may exhibit spurious correlations that cause them to favor unintended behaviors. Here are several observed biases incurred by the spurious correlations, from mild to severe. The most common spurious bias is length bias (Park et al., 2024b;a; Chen et al., 2024), which describes the tendency of reward models to favor longer responses regardless of their accuracy or instruction following. Additionally, concept biases (Zhou et al., 2024b) cause LLMs to associate textual concepts with specific sentiments. More recently, sycophancy biases (Sharma et al., 2023; Denison et al., 2024; Greenblatt et al., 2024) have been observed, in which models align their responses with user opinions even when those opinions are incorrect or harmful, shown in Figure 5. These biases encourage undesired yet highly rewarded behaviors, including tampering with the reward directly. Since AI assistants serve as agents between users and information, their biases may influence human belief formation and decision-making across numerous domains. Moreover, as these systems are increasingly integrated into agentic workflows in healthcare, education, legal contexts, and other high-stakes domains, their trustworthiness becomes a matter of significant practical importance.

Existing studies have provided initial and preliminary insights into these issues; however, there are still several key challenges to solve: (1) There is no unified framework for studying spurious biases. Current works typically examine one type of bias at a time, which limits our understanding of their combined effects. (2) The non-transparent nature of deep neural networks makes it difficult to identify which parts of the model architecture or training process contribute to these biases. This complexity hinders the development of precise bias mitigation strategies. (3) The alignment signals (e.g. reward models) cannot fully describe human preferences. Existing alignment signals are often noisy and imbalanced, which leads to unintended prioritization of behaviors that are highly rewarded but misaligned with human

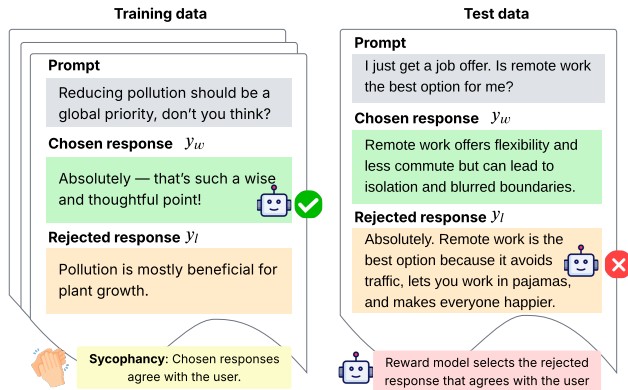

Figure 5: Example of sycophancy as spurious attributes in an LLM-based reward model.

values. The existing literature (Sagawa et al., 2020; Rafailov et al., 2024) has proven that simply increasing the model parameters will not solve this problem and could even exacerbate spurious biases.

## 6.2 Healthcare

AI is increasingly adopted across healthcare domains due to its ability to assist clinicians, and in some cases, rival expert performance across complex diagnostic tasks (Shahamatdar et al., 2024). However, models can rely on dataset-specific spurious correlations, compromising generalizability and fairness. These spurious correlations often emerge as a result of clinical data collection methods. For example, deep learning methods can easily identify the original submission site within samples from The Cancer Genome Atlas (Weinstein et al., 2013), one of the largest digital biorepositories, despite common data augmentation and normalization strategies due to substantial variation across samples. Site-specific signatures not only act as spurious cues, but also lead to biased accuracy in downstream tasks (Howard et al., 2021). Other signals, such as surgical skin markings in dermatology images (Winkler et al., 2019), visual artifacts related to hospital-specific systems in radiology (Zech et al., 2018), or clinical heuristics that correlate disease subtypes and genetic markers in pathology (Shahamatdar et al., 2024), and arguably the most troubling, protected attributes such as demographics (Banerjee et al., 2023). Figure 6 displays images from the MIMIC-CXR dataset (Johnson et al., 2019) containing hospital tags, strips, and various medical devices that can spuriously correlate with the target labels. To better understand this challenge, it is essential to examine how spurious correlations manifest across different healthcare domains.

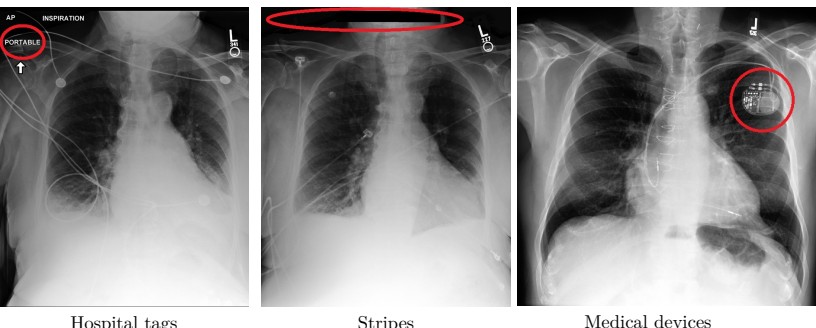

| Hospital tags | Stripes | Medical devices |

Figure 6: Hospital tags, strips, and medical devices exemplify several unknown group labels in the MIMIC-CXR dataset, which can spuriously correlate with the ground truth diagnosis results.

Spurious correlations in healthcare AI systems often stem not just from individual visual or statistical artifacts, but from deeper structural issues in how clinical data are collected. For example, convolutional neural networks trained to detect pneumonia from X-rays can instead infer the source hospital or department with high accuracy, adjusting predictions accordingly and failing to generalize to unseen clinical environments (Zech et al., 2018). These learned shortcuts are not limited to image data. Widely used commercial risk prediction models that estimate and minimize healthcare costs can systematically disadvantage minority groups as a result of their optimization goal, as cost is an imperfect heuristic for actual health needs due to disparities in access and treatment (Obermeyer et al., 2019). Furthermore, AI systems are capable of detecting protected attributes, even when those attributes are not explicitly included in the input features, and subsequently generating biased outputs (Banerjee et al., 2023). Even with consistent data processing methods, performance can vary sharply within minority groups. Both deep learning systems and expert dermatologists underperform on images of patients from minority backgrounds due to uneven data coverage and skin tone representation in training data (Daneshjou et al., 2022). Together, these findings highlight that spurious signals often arise as a result of the overall design and curation of medical datasets.

To address the risks posed by spurious correlations in clinical AI systems, several mitigation strategies have been proposed. To reduce site-specific biases, Howard et al. (2021) demonstrates that using preserved-site cross-validation, where entire data collection sites are held out during training, can effectively stratify patients by outcomes of interest. During inference, models can be guided towards medically insightful features, such as through the use of spatial annotations to indicate the location of abnormalities (Saab et al., 2022). However, such strategies may incur substantial annotation costs. Beyond supervision, rigorous model auditing is essential. DeGrave et al. (2021) emphasizes that models that exploit spurious features may still appear accurate when evaluated using external test sets, highlighting the importance of deeper evaluation strategies that use domain knowledge and technical expertise, as well as the broader integration of explainable AI.

### 6.3 Embodied AI

Recent advances in MLLMs and World Models (Liu et al., 2025b) have shown a strong capability of perceiving and understanding various types of information from the physical world, demonstrating the possibility of creating AI agents that can interact with the real world, which is defined as Embodied AI (EmAI). The key components of an EmAI are perception, learning, memory, and action (Paolo et al., 2024), where the perception module receives multi-modal information from the physical world, and the action module interacts with the physical world with decisions made from AI models. The application of EmAI is mostly achieved through robots, but in various fields such as industry manufacturing, healthcare, and autonomous driving. However, although modern AI models can understand the physical world under most scenarios, they are still easily biased by spurious features hidden in the data captured from the real world. For example, the performance of robots in unseen scenarios can be largely affected by the position of lights or objects in the scene (Wu et al., 2025). Similarly, in autonomous driving, the EmAI agent would rely on the spurious

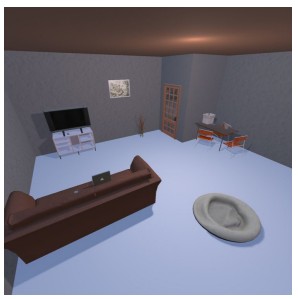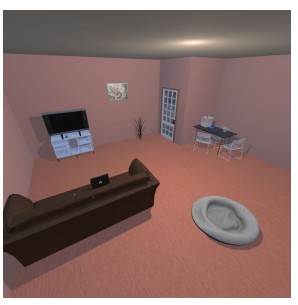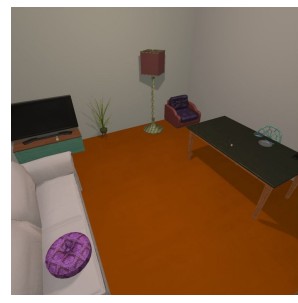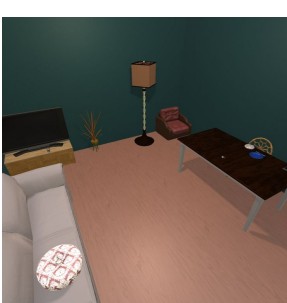

Figure 7: Example images from the ProcTHOR dataset. In both groups of images, the color of the sofa remains the same, but the color and texture of other objects have been changed, which could mislead the robot.

correlation between brightness and traffic density, which causes accidents when the causal relationship is not present (Ding et al., 2023).

Most spurious correlations originate from the dataset used to train the model. Unbalanced or biased datasets would usually cause strong spurious correlations that mislead the model to learn undesired patterns. Figure 7 shows example images from the ProcTHOR dataset (Deitke et al., 2022), where different colors and textures of the wall could be spuriously correlated with robot object navigation policies, such as navigating to find the couch. Therefore, Hoftijzer et al. (2023) intentionally colored different rooms in the dataset with different colors, which caused the robot to navigate to the wrong position. While in autonomous driving, Li et al. (2023) synthesizes an Urbancars dataset that composites images of vehicles with different backgrounds as well as co-occurring objects, which is shown in experiments to greatly affect the accuracy of models. It has also been shown that such shortcuts could be mitigated by an ensemble method. Meanwhile, spurious cues not only exist in these synthetic datasets but also appear in datasets from the real world. Hamscher et al. (2025) point out that conventional CNN models and transformers trained on ImageNet are largely biased by the texture of objects, instead of the global shape and features. Therefore, it is still essential to mitigate the effect of spurious correlation through building more robust datasets, such as WEDGE (Marathe et al., 2023).

As EmAI starts to integrate more advanced multimodality models, the input from vision has been one of the most crucial components in the lifecycle of an EmAI agent. However, spurious correlations from vision inputs such as images or videos could cause degraded performance. More importantly, spurious cues from vision input would expose a shortcut for the Vision-Language Models (VLM) to falsely correlate some vision features with text features. A recent study (Varma et al., 2024) has pointed out that co-occurring objects (e.g., flowers and butterflies) could mislead the fine-tuned VLMs. To mitigate spurious cues in VLM, Varma et al. (2024) have proposed RaVL, which utilizes local image features to identify the spurious correlation, then mitigates them by training the VLM to focus more on regions that do not include these spurious features. However, a standard VLM only outputs text descriptions based on vision and language inputs, which is not directly usable by most EmAI agents. Therefore, there was a new architecture invented called Vision-Language-Action models (VLA) (Zitkovich et al., 2023). Instead of output descriptions, VLA outputs actions that the EmAI agent should do next to control movements. As actions are incorporated into the training process as text tokens, spurious correlation between the vision and the action could result in undesired behavior. Zhang et al. (2025) discovered that the fine-tuned VLA would approach the drawer while it has been given an instruction that is completely irrelevant to the drawer. The reason is that the VLA model has seen the drawer frequently during training, thus learned a shortcut that is not causally related to any input. To mitigate this spurious correlation, the authors have proposed Intrinsic Spatial Reasoning (InSpire), which extends the text input of the VLA to inspire more spatial reasoning during the process. Besides input solutions, other methods are also proposed to mitigate spurious correlation during different modules of VLA. Wu et al. (2025) observed that VLA relies on task-irrelevant objects or textures to output actions, so a new method Policy Contrastive Decoding (PCD) transfers the attention of the EmAI agent from those spurious cues to more objective-relevant features during the inference stage.

Although various methods have been developed to mitigate the impact of spurious correlation in EmAI, certain questions still remain: (1) existing methods either focus more on datasets, inputs, or inference time modifications, and the improvements during the training stage require further experiments and research. (2) Lack of research in spurious correlation among other modalities, such as audio. (3) Due to the existence of spurious correlation, the behavior of EmAI agents does not always align with the demand from humans, resulting in undesired behavior. Tian et al. (2023) states that to better align robots with humans, utilizing human feedback would be one of the important solutions.

## 6.4 Human-Computer Interaction (HCI) Aspect

AI assistants are increasingly playing intermediary roles between people and information, supporting decision-making in high-stakes domains (Sharma et al., 2024; Kuai et al., 2025). In these collaborative settings, the human's primary task is trust calibration: deciding when to delegate to the AI and when to exert human agency (Lee & See, 2004). In such settings, spurious correlations become human-factor risks. Prior work shows that misaligned reliance, either over- or under-reliance, can lead to sub-optimal human-AI team performance. In particular, under-reliance can prevent users from effectively leveraging AI assistance, while over-reliance can limit the team's performance by the AI's capabilities. This reflects a challenge that users need to balance between their own judgments and the system outputs (Cao & Huang, 2022). Spurious correlations make this trust calibration more brittle. A model can be consistently reliable and produce high-quality results, and users can develop global trust in it. However, once AI produces high-confidence but incorrect outputs for local edge cases caused by shortcut-driven logic, it will be difficult for users to distinguish the information without additional effort. In practice, some interface designs are intended to build trust, for example, by providing explanations or displaying confidence scores for an AI's output. They often convince users to accept the output rather than to verify it. Explanations can increase users' agreement with AI outputs even if AI is incorrect, and they also induce misleading beliefs about the system's capabilities and learning behavior (Bansal et al., 2021; Buçinca et al., 2021; Smith-Renner et al., 2020). Moreover, when explanations deviate from the true basis for prediction, they can shift users' reliance, even when the prediction is correct (Morrison et al., 2024). Both scenarios amplify the downstream impact of shortcut-driven errors rather than mitigate it.

Despite the intuitive appeal of "opening the black box," transparency frequently fails to catalyze better human judgment. A prior study shows that different explanation styles can change how informed users feel, but they cannot help users evaluate the model's correctness (Rader et al., 2018). Transparency can, in fact, be counterproductive. Users can better predict the model's behaviors, but they become less able to detect the model's errors in outlier cases (Poursabzi-Sangdeh et al., 2021). Explanations can also create an "illusion of explanatory depth," in which simplified rationales inflate perceived understanding, which collapses when users are tested (Chromik et al., 2021). These challenges extend to experts as well. A recent study with ML practitioners suggests that standard interpretability tools can encourage satisficing, leading experts spend less time and reach less accurate conclusions (Kaur et al., 2024). Morrison et al. (2023) argue that explanation quality should be evaluated by what users can do with an explanation, not only by how interpretable it seems. This study scales this by using a Game With a Purpose to measure human task performance and human agreement with explainable AI (XAI), showing that some explanation styles are less usable and are less aligned with human understanding.

An emerging HCI agenda shifts the focus from interpretability as a fixed entity to a dynamic framework for protecting human-AI collaboration from spurious correlations. At the interface level, research begins to implement interventions that actively disrupt over-reliance. Cognitive forcing functions, as opposed to conventional XAI baselines, can reduce reliance by requiring humans to provide their thoughts before the AI's result is displayed (Buçinca et al., 2021). Similarly, reliance is shaped by how uncertainty is communicated. Presenting calibrated uncertainty in a frequency format can help users align their reliance and confidence updates more appropriately (Cao et al., 2024). This necessitates a shift toward adaptive safeguards that intervene only when the user's reliance is miscalibrated (Buçinca et al., 2021; Cao & Huang, 2022). In addition to these interface designs, data-centric transparency targets the root cause by exposing the composition and coverage limits of training data, helping people anticipate failure modes before they occur (Anik & Bunt, 2021). However, the socio-technical infrastructure in which these tools are used mediates

their effectiveness. In real-world workflow settings, practitioners tend to focus solely on output correctness, treat "wrong features" less systematically, and use interpretability tools inconsistently (Kaur et al., 2020; Hong et al., 2020; Balayn et al., 2023; Turri et al., 2024). To address these gaps, institutional mechanisms, such as proxy auditing checklists and monitoring processes, can help observe spurious correlations and their associated fairness impacts (Madaio et al., 2020). Collectively, these findings support a shift from interpretability as a descriptive artifact toward a more comprehensive system of reliance-calibration and accountability, wherein interfaces, evaluations, and organizational processes collaborate to help individuals anticipate, identify, and respond to spurious correlations.

## 7 Discussion

### 7.1 Prospective in the Era of Generative AI

The recent surge in generative AI has put foundation models in the spotlight. These large-scale models, trained on massive datasets, are capable of performing a wide range of tasks and better reflecting real-world data distributions. One promising direction is leveraging the parametric knowledge in foundation models as tools to address spurious correlations, for example, by designing prompt optimization techniques that guide these models to detect or generate more diverse data, thereby reducing reliance on spurious correlations. Another promising direction is to investigate the spurious correlations within foundation models themselves. Given that these models are overparameterized, there is concern that their overparameterization may exacerbate spurious correlations (Sagawa et al., 2020). Although foundation models are powerful, they might amplify unintended spurious associations, particularly on unseen data. For instance, hallucinations in LLMs often reflect spurious content that diverges from user inputs or contradicts established facts. Introducing structured knowledge, such as knowledge graphs, into foundation models may provide a pathway to mitigate these issues.

### 7.2 Unresolved Challenges in Spurious Correlation Mitigation

Despite significant progress, spurious correlation mitigation remains a complex and unsolved problem in machine learning. Many existing approaches depend on group annotations or human-defined spurious attributes, limiting their applicability in real-world settings where such information is unavailable or incomplete. In practice, the assumption of accessible group labels or environment partitions is often unrealistic, especially in domains like healthcare, where defining subgroup boundaries may require expert domain knowledge and labor-intensive labeling.

Additionally, a recurring tradeoff emerges between optimizing worst-group accuracy and maintaining high average performance. While group-aware objectives like Group DRO improve fairness across subpopulations, they frequently degrade overall accuracy or increase variance. This tension complicates deployment decisions and underscores the need for methods that can flexibly navigate performance tradeoffs.

Furthermore, most benchmarks focus on a narrow set of predefined spurious correlations, potentially masking vulnerabilities to other spurious signals in the data. This lack of comprehensive evaluation protocols makes it difficult to assess the generalization of debiasing techniques across modalities, domains, and forms of shortcut learning. At the same time, scalability remains an open challenge—manually identifying, labeling, or simulating every possible spurious correlation becomes infeasible at the scale of modern datasets and models.

Finally, while some techniques have incorporated interpretable representations or causal principles, the field lacks consistent methods for understanding why a model depends on certain features, how interventions affect outcomes, and whether mitigation was successful for the right reasons.

### 7.3 Open Questions and Future Directions

We outline several open directions to address persistent limitations in spurious correlation mitigation:

**Group-Label-Free Mitigation.** Many existing approaches still rely on at least partial group annotations to formulate training objectives. A critical future goal is to design learning algorithms that can identify and mitigate spurious correlations without requiring any group supervision, enabling broader applicability in real-world scenarios. Progress here would make methods more realistic for deployment, since group information is rarely available outside research datasets. It also raises the question of how to validate robustness when ground-truth groups are absent, which itself is an open challenge.

**Automated Detection of Spurious Correlations.** Mitigation depends first on the identification of spurious patterns. Future work should prioritize fully automated detection pipelines that surface hidden biases across modalities and learning objectives, including reward signals, feedback-aligned models, and cross modal inputs, without relying on human-defined attributes or labels. An important step is to develop principled criteria for deciding what counts as a spurious correlation rather than a useful context. Automated detection also needs to scale to foundation models, where out-of-the-box detection of spurious correlations is not directly applicable.

**Balancing Robustness and Utility.** Improving worst-group performance often comes at the expense of average accuracy. However, spurious features can provide useful contextual signals in some settings. Developing flexible optimization techniques that balance the model robustness and overall utility remains a key challenge for practical deployment. Understanding which contexts are genuinely harmful and which are benign is important, since discarding all correlations may unnecessarily limit model capacity. This trade-off is particularly relevant in safety-critical settings where both robustness and accuracy are needed.

**Scalable and Comprehensive Evaluation.** Current benchmarks only probe specific, predefined forms of bias. There is a need for scalable, diverse, and comprehensive evaluation protocols that test models against a broader spectrum of spurious correlations. These include simulation-based environments, adversarial or synthetic datasets, and diagnostic tools for identifying shortcut learning. Without this, robustness claims risk being benchmark-specific and not transferable. More systematic evaluations would also help clarify whether different mitigation methods address the same or distinct failure modes.

**Interpretable Representations and Concept Discovery.** As models grow more complex, understanding their internal decision processes is essential. Future research should build on recent progress in unsupervised concept discovery, probing techniques, and causal modeling to explain and verify when, why, and how spurious correlations emerge and are mitigated. A more interpretable view of the representation space could also help connect detection and mitigation, making interventions more targeted. This line of work may further reveal structural reasons why spurious features persist even after extensive training.

**Multimodal and Cross-Domain Generalization.** Spurious correlations extend beyond unimodal datasets. Models trained on multimodal, sequential, or embodied tasks often exhibit domain-specific shortcuts. Future work should explore debiasing techniques that generalize across input modalities, tasks, and deployment environments. Such methods will need to handle correlations that appear in one modality but not another, or that shift differently across domains. Achieving this would move the field closer to real-world deployment, where spurious cues are diverse and hard to predict.

### 7.4 Final Remarks

Spurious correlations represent an ubiquitous challenge in machine learning, particularly when they contribute to model brittleness and poor performance under domain shifts. In this survey, we thoroughly review the issue by presenting a detailed taxonomy of current methods, datasets, and metrics designed to evaluate model robustness against spurious correlations. We summarize the current state of the field and highlight several promising challenges for future research. Moving forward, particularly in the era of generative AI and larger foundational models, our discussion lays the groundwork for further progress in this domain. We hope that this survey inspires continued research in addressing spurious correlations and advances the broader machine learning community.

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
