# OpenReview forum: "The Clever Hans Mirage: A Comprehensive Survey on Spurious Correlations in Machine Learning"
_TMLR — Accepted by TMLR_

### Review · Reviewer_7DcQ · 2025-11-04

**Summary Of Contributions:**

**Summary**

This survey on spurious correlations uses the Clever Hans metaphor to motivate the problem, formalizes a group-based setup `g=(y,a)` with core metrics (worst-group, average-group, bias-conflicting), and explains why models latch onto shortcuts (simplicity bias, training dynamics). It organizes methods into four families (data-centric, representation-centric, post hoc, scenario-specific) with practical notes on assumptions and costs, consolidates benchmarks and evaluation practices across CV, NLP, Graph, Multimodal, RL, and Healthcare, and outlines key open directions such as label-free detection and mitigation, robustness vs utility trade-offs, scalable evaluation, and cross-modal causal grounding.

**Key Strengths**
1) Clear threat model and notation
2) Comprehensive and timely taxonomy, including LLM alignment biases such as sycophancy and length bias
3) Broad cross-domain coverage (CV, NLP, Graph, Multimodal, RL, Healthcare)
4) Pedagogically useful figures and tables

**Weaknesses**
1) Taxonomy is clear but not yet an operational framework. A compact decision map tying assumptions (group labels, supervision strength, train vs test adaptation) to method families would improve practitioner usability.
2) Limited quantitative aggregation. A lightweight meta-analysis summarizing typical changes in worst-group accuracy and overall accuracy on common benchmarks would strengthen evidence.

**Audience:**

Yes

**Audience Explanation:**

TMLR’s audience actively works on distribution shift, robustness, and fairness. This survey offers a clear synthesis across CV, NLP, Graph, Multimodal, RL, and Healthcare, including timely topics like LLM alignment biases. The taxonomy, metrics, and practical guidance make it useful for both researchers and practitioners.

**Broader Impact Concerns:**

**Broader Impact Concerns**

- Debiasing guidance may shift utility across groups, creating new disparities or lowering overall accuracy.
- Post-hoc or test-time adjustments may overfit proxy signals for group labels and worsen calibration.
- Collecting or inferring group attributes may raise privacy and compliance risks in real-world datasets.

**Claims And Evidence:**

Yes

**Claims Explanation:**

The claims are conceptual syntheses grounded in well-cited prior work, with accurate definitions of the setup and metrics. Coverage across CV, NLP, Graph, Multimodal, RL, and Healthcare is clear, and figures/tables support the taxonomy. As a survey, it summarizes published results rather than adding new experiments, which is appropriate. A brief meta-analysis would strengthen the evidence, but overall support is adequate.

**Requested Changes:**

1) Add an operational overview figure and a short decision map that link assumptions to method families. Keep it compact so readers can choose methods by: group labels available or not, supervision level, and train vs test-time adaptation.

2) Include a lightweight quantitative summary on a few common benchmarks. A small table with typical changes in worst-group and overall accuracy by method family is enough to ground the main takeaways.

---

> ### Author Response · Authors · 2025-12-10
>
> Thank you for your constructive comments on our submission. We agree that the existing taxonomy map is not that intuitive for researchers to choose methods based on their experiment settings. We have added a compact decision map in Section 4 Figure 4.
>
> In addition, we have added a lightweight table, Table 2 in Section 5, to bring more quantitative results from existing literature. It covers the worst group and average accuracy over several common datasets as requested, which allows readers to get a quick view of performance between different representative methods.

---

### Review · Reviewer_sBvv · 2025-11-07

**Summary Of Contributions:**

This paper provides a broad survey on spurious correlations in machine learning. The topic is important and timely, and the taxonomy attempts to organize a large and rapidly growing literature. However, the paper suffers from several issues in conceptual clarity, mathematical rigor, and notational consistency, which significantly limit its usefulness as a reliable reference.

**Audience:**

Yes

**Audience Explanation:**

NA

**Claims And Evidence:**

No

**Claims Explanation:**

1. Definition 2.1 is conceptually mathematically unclear. The spurious correlation is defined by introducing a non-standard and mathematically undefined mapping $\phi: \mathcal A \to \mathcal Y^{K'}$, whose meaning, role, and necessity are not explained. It is also unclear how this mapping relates to the spurious correlation $\langle y, a\rangle$. Furthermore, several notational choices raise questions: What exactly is $\mathcal Y^{K'}$, a Cartesian product of label sets? What is the nature of the attribute $a$? Is $a$ assumed to lie in the same space or dimension as $x$, or are they structurally different? As written, the definition is not informative and does not rigorously or coherently capture the intended notion of spurious correlation.

2. Theoretical formulation in Section 2.3 is not rigorous. Equation (3) introduces expressions such as: $\langle \phi(x), w_y\rangle
\approx  \langle \phi(a), w_y\rangle \gg  \langle \phi(x_c), w_y\rangle.$ Deep feature extractor $\phi$ do not naturally decompose representations into independent components of the input. Thus the quantities $\phi(a)$ and $\phi(x_c)$ are not well-defined, i.e., $\phi$ is a function taking input $x$ and one has the decomposition $x = (x_c, a)$. Then,  $\phi(a)$ and $\phi(x_c)$ are not correct mathematically.  Furthermore, the inequality has no theoretical justification and the expression implicitly assumes that representations decompose additively or linearly, which is not valid for deep nonlinear models.

3. The symbol $\phi$ and $\langle \cdot, \cdot\rangle$ is overloaded. In Definition 2.1, $\phi$ denotes a strange one-to-many mapping from attributes to class subsets and $\langle y, a\rangle$ is used as a notation for the spurious correlation. However, in Section 2.3, $\phi$ suddenly denotes the neural network’s feature extractor and $\langle \cdot, \cdot\rangle$ denotes the standard inner product. These two meanings are completely unrelated. Using the same symbol creates unnecessary confusion and reflects a lack of notational discipline.

4. The ERM formulation contains probabilistic inconsistencies. The paper writes that a sample $(x,y,a)$ is drawn “conditioned on group label $g=(y,a)$.” However, conditioning on $g$ fixes both $y$ and $a$; only $x$ should be random under $P_{x \mid g}$.   Thus, writing “draw $(x,y,a)$” from such a conditional distribution is mathematically incorrect and may confuse readers unfamiliar with group-based formulations.

**Requested Changes:**

NA

---

> ### Author Response · Authors · 2025-12-10
>
> We thank the reviewer for the detailed comments. We would like to clarify that Section 2 is not intended to introduce a new theoretical analysis, but to summarize and standardize definitions and intuitions that are already widely used in the spurious correlation and group robustness literatures, e.g., [1,2,3], given it is a survey paper submission. Below we address each concern and revised the draft.
>
> 1. The definition follows the standard group-based formulation used in prior works on spurious correlation and worst-group robustness, where spurious correlations are operationalized through the joint occurrence of target labels $y$ and spurious attributes $a$, and groups are defined as $g=(y,a)$. The notation $\langle y, a\rangle$ is used only as a shorthand for such co-occurrences, not as a mathematical operator. The main purpose of Definition 2.1 is to formalize dataset-induced group structure rather than a causal or functional relationship. $\mathcal{Y}^{K^{\prime}}$ means the label set with $K^{\prime}$ classes. We add the clarification in the revised paper.
> 2. The original expression in Equation 3 was intended as a conceptual illustration linking empirical findings on learning dynamics with theoretical intuitions about early reliance on spurious features. To avoid unintended mathematical implications and to present the intuition more clearly, we have removed the equation and revised the text to provide a more clear description. This revision clarifies the intention of this subsection.
> 3. Thank you for highlighting this issue. The revised manuscript removes overloaded notation, including the earlier use of angle brackets and $\phi(a)$ to avoid confusions.
> 4. In the revision, we clarify that samples are drawn as $x \sim P(x \mid g)$, consistent with standard ERM and Group DRO formulations, and revise the text to avoid suggesting that $(x,y,a)$ are jointly random after conditioning.
>
> [1] An Investigation of Why Overparameterization Exacerbates Spurious Correlations, ICML 2020 \
> [2] Change is Hard: A Closer Look at Subpopulation Shift, ICML 2023 \
> [3] Environment Inference for Invariant Learning, ICML 2021

---

### Review · Reviewer_J6Fz · 2025-11-26

**Summary Of Contributions:**

This paper provides a comprehensive survey of *spurious correlations* in machine learning. It includes the definitions of spurious correlations, theoretical insights, categorization of mitigation methods, datasets, and evaluation metrics. It also discusses the impacts in different domains, such as large language models (LLMs), healthcare, and embodied AI. The authors divide existing approaches into four categories: data-centric methods, representation learning, post-hoc methods, and specialized methods. The survey also highlights the challenges posed by spurious correlations in modern foundation models and discusses future research directions.



**Strengths**

- Very broad and thorough coverage of literature across multiple ML domains (computer vision, healthcare, LLMs, robotics, and so on).
- Clear high-level structure and well-organized taxonomy (Figure 3).
- Useful consolidation of benchmark datasets and evaluation metrics.

**Weaknesses**

- Analysis remains mostly descriptive. Deeper meta-analysis is required. For example, analyzing the method failures or discussing the trade-offs among different kinds of methods.
- Some important concepts require more precise definitions. For example, in Equation 3, more explanation for core features $x_c$ is required: What are the core features $x_c$? Is it the real feature that causes y?
- Font size in Figure 4 is related small.

**Audience:**

Yes

**Audience Explanation:**

Spurious correlations are related to many research areas covered by TMLR.

**Broader Impact Concerns:**

This paper does not have ethical concerns.

**Claims And Evidence:**

Yes

**Claims Explanation:**

The paper provides a large number of references and accurately summarizes prior works.

**Requested Changes:**

1. The definition of core features $x_c$ needs to be clarified. It is currently unclear whether $x_c$ refers to the features caused by $y$, the invariant features, or simply the non-spurious part of $x$. A more explicit and formal explanation would improve clarity.
2. Adding more discussion on failure cases or trade-offs among different methods.

---

> ### Author Response · Authors · 2025-12-10
>
> We thank the reviewer for their constructive comments. We have revised the manuscript accordingly and summarize key changes below.
>
> 1. We appreciate the suggestion and thank the reviewer for pointing out this ambiguity. We have revised section 2.3 to provide a precise definition of $x_c$ to make clear that core features are the invariant/causal components of the input rather than just the non-spurious residual.
>
> 2. We appreciate the feedback and also see clear opportunities to strengthen the survey by providing deeper comparitive insights across each methodological category. In response, we have added a "Discussion" component following each major subsection within the Methodology that analyzes common failure cases, assumptions, and trade-offs within each strategy. These additions provide a substantially deeper meta-analysis of the surveyed methods.
>
> 3. We have changed the size of Figure 4 in the revision to make it clearer to read.

---

### Author Response · Authors · 2025-12-10
**General Response to the Reviewers**

Dear Reviewers,

Thank you for the constructive comments and suggestions on our manuscript. They have helped us improve the clarity and completeness of the survey. We have revised the paper accordingly, and all changes are reflected in the updated PDF on the submission page. We appreciate any further feedback.

Best regards, \
Authors of Submission 6058

---

### Decision · Action_Editor_1J5q · 2026-01-20

**Recommendation:** Accept with minor revision

**Additional Comments:**

This paper presents a comprehensive survey of spurious correlations in machine learning. It introduces a taxonomy of mitigation methods (data-centric, representation learning, post-hoc, and specialized methods), summarizes key benchmarks and metrics, and discusses implications for modern foundation models.

Reviewer sBvv raised significant concerns regarding the mathematical rigor of the initial definitions (specifically Definition 2.1 and Equation 3). The authors have addressed this by removing the controversial Equation 3 to avoid misleading theoretical implications and clarifying the notation for dataset-induced group structures. While Reviewer sBvv retained a "Leaning Reject," the specific technical errors identified appear to have been resolved in the revision.

To ensure the survey serves as a reliable reference, a comprehensive revision is required regarding mathematical rigor and clarity. The authors should conduct a careful and comprehensive revision of the entire manuscript, including:

- Notation Consistency: Ensure strict consistency in mathematical notation throughout the paper. Verify that symbols  are defined once and not overloaded with conflicting meanings across different sections.

- Equation Rigor: Review all remaining equations to ensure they are mathematically precise and necessary.

- Logical Flow: Ensure a smooth logical progression between the formal problem definitions in Section 2 and the categorization of methods in Section 4. The connection between the theoretical formulation of the problem and the proposed solutions should be seamless.

**Audience:**

Yes

**Audience Explanation:**

The topic of spurious correlations and out-of-distribution robustness is of central importance to the TMLR audience. Given the rapid proliferation of methods in this space, a consolidated survey that organizes benchmarks, metrics, and mitigation strategies is highly relevant for both practitioners and researchers looking for a structured entry point into the field.

**Claims And Evidence:**

Yes

**Claims Explanation:**

As a survey paper, the "evidence" consists of an accurate and comprehensive synthesis of the existing literature. The authors have supported their taxonomy and analysis by referencing a wide range of works across computer vision, NLP, and healthcare.